# A conserved function for pericentromeric satellite DNA

**Madhav Jagannathan[1], Ryan Cummings[1,3], Yukiko M Yamashita[1,2,3]***

[1]Life Sciences Institute, University of Michigan, Ann Arbor, United States;
[2]Department of Cell and Developmental Biology, University of Michigan, Ann Arbor, United States; [3]Howard Hughes Medical Institute, University of Michigan, Ann Arbor, United States

**Abstract** A universal and unquestioned characteristic of eukaryotic cells is that the genome is divided into multiple chromosomes and encapsulated in a single nucleus. However, the underlying mechanism to ensure such a configuration is unknown. Here, we provide evidence that pericentromeric satellite DNA, which is often regarded as junk, is a critical constituent of the chromosome, allowing the packaging of all chromosomes into a single nucleus. We show that the multi-AT-hook satellite DNA-binding proteins, *Drosophila melanogaster* D1 and mouse HMGA1, play an evolutionarily conserved role in bundling pericentromeric satellite DNA from heterologous chromosomes into 'chromocenters', a cytological association of pericentromeric heterochromatin. Defective chromocenter formation leads to micronuclei formation due to budding from the interphase nucleus, DNA damage and cell death. We propose that chromocenter and satellite DNA serve a fundamental role in encapsulating the full complement of the genome within a single nucleus, the universal characteristic of eukaryotic cells.
DOI: https://doi.org/10.7554/eLife.34122.001

*For correspondence:
yukikomy@umich.edu

## Introduction

Satellite DNA is AT-rich, non-coding, repetitive DNA that is abundant in centromeric and pericentromeric heterochromatin. Unlike the satellite DNAs that comprise the vast majority of natural centromeres (*Willard, 1990*; *Sun et al., 1997*, *2003*), the role of pericentromeric satellite DNA remains obscure: although function for a few satellite DNA repeats has been implied in certain cellular processes such as meiotic segregation of achiasmatic chromosomes, X chromosome dosage compensation and formation of lampbrush-like loops on the Y chromosome during male meiosis (*Yunis and Yasmineh, 1971*; *Bonaccorsi et al., 1990*; *Dernburg et al., 1996*; *Menon et al., 2014*), a unifying theme for pericentromeric satellite DNA function remains elusive. Moreover, highly divergent satellite DNA sequences even among closely related species has led to the idea that satellite DNA does not serve a conserved function and is mostly a selfish element or junk (*Doolittle and Sapienza, 1980*; *Walker, 1971*). Pericentromeric satellite DNA repeats are proposed to be sources of genomic instability, as their misexpression is associated with the formation of genotoxic R-loops and DNA damage (*Zhu et al., 2011*; *Zeller et al., 2016*; *Zeller and Gasser, 2017*). Most studies on pericentromeric heterochromatin have focused on the mechanisms to repress satellite DNA transcription, and accordingly, a clear rationale for the existence of most pericentromeric satellite DNA is still lacking.

Cytologically, it is well documented that pericentromeric satellite DNA from multiple chromosomes is clustered into chromocenters in interphase nuclei in diverse eukaryotes including *Drosophila*, mouse and plants (*Figure 1A*) (*Jones, 1970*; *Pardue and Gall, 1970*; *Gall et al., 1971*; *Fransz et al., 2002*). While multiple factors such as epigenetic modifications and transcription of repetitive DNA from pericentromeric DNA sequences are known to be required for chromocenter

**eLife digest** On Earth, life is divided into three domains. The smallest of these domains includes all the creatures, from sunflowers to yeasts to humans, that have the genetic information within their cells encased in a structure known as the nucleus. The genomes of these organisms are formed of long pieces of DNA, called chromosomes, which are packaged tightly and then unpackaged every time the cell divides. When a cell is not dividing, the chromosomes in the nucleus are loosely bundled up together.

It is well known that DNA is the blueprint for the building blocks of life, but actually most of the genetic information in a cell codes for nothing, and has unknown roles. An example of such 'junk DNA' is pericentrometric satellite DNA, small repetitive sequences found on all chromosomes.

However, new experiments by Jagannathan et al. show that, in the nucleus of animal cells, certain DNA binding proteins make chromosomes huddle together by attaching to multiple pericentrometric satellite DNA sequences on different chromosomes. In fact, if these proteins are removed from mice and fruit flies cells grown in the laboratory, the chromosomes cannot be clustered together. Instead, they 'float away', and the membranes of the nucleus get damaged, possibly buckling under the pressure of the unorganized DNA.

These events damage the genetic information, which can lead to the cell dying or forming tumors. 'Junk DNA' therefore appears to participate in fundamental cellular processes across species, a result that opens up several new lines of research.

DOI: https://doi.org/10.7554/eLife.34122.002

formation (*Peters et al., 2001*; *Probst et al., 2010*; *Bulut-Karslioglu et al., 2012*; *Pinheiro et al., 2012*; *Hahn et al., 2013*), the ultimate consequences of disrupted chromocenter formation has never been addressed, leaving the function of chromocenters unknown.

In this study, we explored the role of pericentromeric satellite DNA/chromocenters by studying multi-AT-hook proteins, D1 from *Drosophila melanogaster* and HMGA1 from mouse. D1 and HMGA1 are known to bind specific pericentromeric satellite DNA, and we show that these proteins are required for chromocenter formation. When chromocenters are disrupted in the absence of these proteins, cells exhibited a high frequency of micronuclei formation, leading to DNA breakage and cell death. We show that micronuclei are formed during interphase by budding from the nucleus. We further show that D1 binding to the target DNA sequence is sufficient to bring it to the chromocenter. High-resolution imaging revealed chromatin threads positive for D1/HMGA proteins and satellite DNA that connect heterologous chromosomes. Taken together, we propose that chromocenter formation via bundling of satellite DNA from heterologous chromosomes functions as a mechanism to encapsulate the full complement of the genome into a single nucleus. We suggest that satellite DNA function as a critical constituent of chromosomes and may serve an evolutionarily conserved role across eukaryotic species.

## Results

### The multi-AT-hook satellite DNA binding proteins, *Drosophila* D1 and mouse HMGA1, localize to chromocenters

D1 in *Drosophila melanogaster* and HMGA1 in mouse are multi-AT-hook proteins; D1 contains 10 AT-hooks, HMGA1 contains three AT-hooks and both proteins contain C-terminal acidic domains (*Aulner et al., 2002*). D1 and HMGA1 are known to bind the *Drosophila* $\{AATAT\}_n$ satellite DNA (~8% of the *Drosophila* male diploid genome) and mouse major satellite DNA (~6% of the mouse genome), respectively (*Goodwin et al., 1973*; *Rodriguez Alfageme et al., 1980*; *Levinger and Varshavsky, 1982b*; *Levinger and Varshavsky, 1982a*; *Lund et al., 1983*). The $\{AATAT\}_n$ satellite is distributed across 11 loci on multiple chromosomes as visualized by DNA fluorescence in situ hybridization (FISH) on mitotic chromosome spreads (*Figure 1B*) (*Lohe et al., 1993*; *Jagannathan et al., 2017*). However, it is typically clustered into a few foci in *Drosophila* interphase nuclei, colocalizing with the D1 protein (*Figure 1C*). The D1/$\{AATAT\}_n$ foci stained positively for H3K9me2 in interphase nuclei (*Figure 1C*), a well-established characteristic of constitutive

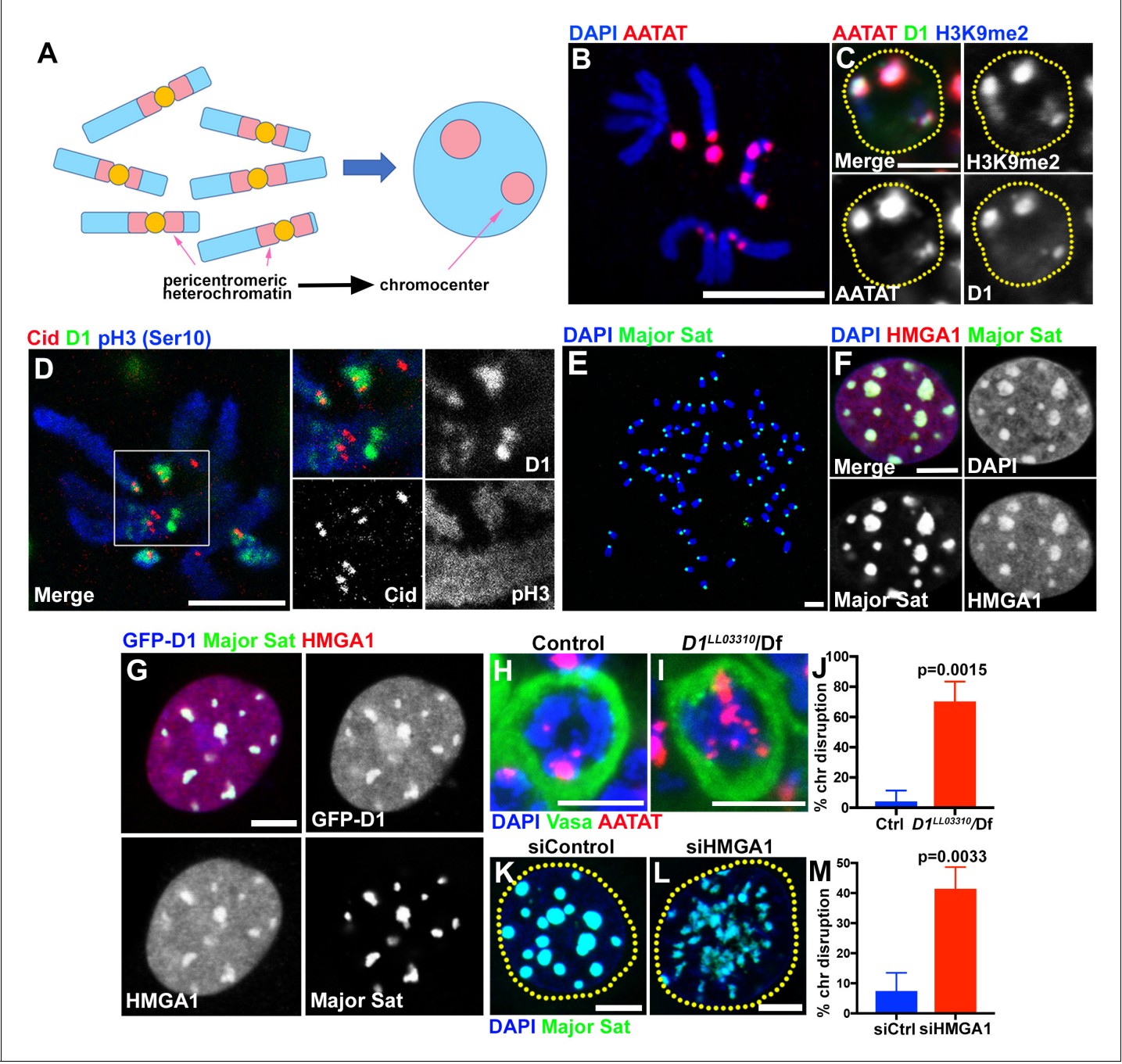

**Figure 1.** Multi-AT-hook proteins, D1 and HMGA1, are required for chromocenter formation in *Drosophila* and mouse cells. (A) Schematic of pericentromeric heterochromatin being organized into the chromocenter. (B) FISH against {AATAT}$_n$ satellite (red) on the *Drosophila* neuroblast mitotic chromosomes co-stained with DAPI (blue) indicating the location of {AATAT}$_n$ in the *Drosophila* genome. (C) FISH against {AATAT}$_n$ satellite (red) in spermatogonial cells immunostained for H3K9me2 (blue) and D1 (green). Dotted lines indicate nucleus. Bars: 5 μm. (D) *Drosophila* neuroblast mitotic chromosomes stained for D1 (green), phospho-histone H3 Serine 10 (pH3-S10) (blue) and Cid/CENP-A (red). (E–G) FISH against the mouse major satellite (green) on C2C12 mitotic chromosomes co-stained with DAPI (blue) (E), in interphase MOVAS cells co-stained for DAPI (blue) and HMGA1 (red) (F) and in MOVAS cells expressing GFP-D1 (blue) stained for HMGA1 (red) (G). (H, I) FISH against {AATAT}$_n$ satellite (red) in control (*D1^{LL03310}*/+) (H) and *D1^{LL03310}*/Df (I) spermatogonial cells stained for DAPI (blue) and Vasa (green). (J) Quantification of spermatogonial cells with disrupted chromocenters (+/+ control n = 117, *D1^{LL03310}*/Df n = 89) from three independent experiments. p-Value from student's t-test is shown. Error bars: SD. (K, L) FISH against the major satellite (green) in siControl (K) and siHMGA1 (L) transfected MOVAS cells co-stained with DAPI (blue). (M) Quantification of cells with disrupted chromocenters from siControl (n = 304) and siHMGA1 (n = 329) from three independent experiments.

DOI: https://doi.org/10.7554/eLife.34122.003

*Figure 1 continued on next page*

*Figure 1 continued*

The following figure supplements are available for figure 1:

**Figure supplement 1.** Multi-AT-hook proteins, *Drosophila* D1 and mouse HMGA1, localize to chromocenters in various mouse cell types.

DOI: https://doi.org/10.7554/eLife.34122.004

**Figure supplement 2.** *Drosophila* D1 and mouse HMGA1 are required for chromocenter formation.

DOI: https://doi.org/10.7554/eLife.34122.005

heterochromatin/chromocenters (*Guenatri et al., 2004*). Consistently, D1 localized near the centromere (marked by *Drosophila* CENP-A, Cid) on mitotic chromosome spreads (marked by phospho-H3 S10) (*Figure 1D*). These results suggest that D1 is a chromocenter-localizing protein, via its binding to the {AATAT}$_n$ satellite DNA.

The mouse HMGA1 protein was originally identified as an abundant non-histone component of mammalian chromatin (*Goodwin et al., 1973*; *Lund et al., 1983*) with subsequent studies demonstrating its binding to satellite DNA (*Strauss and Varshavsky, 1984*; *Radic et al., 1992*). Mouse major satellite, which is present in pericentromeric regions of all chromosomes (*Figure 1E*) (*Lyon and Searle, 1989*), clustered into DAPI-dense chromocenters positive for HMGA1 protein (*Figure 1F*, *Figure 1—figure supplement 1A,B*), revealing an analogous relationship to D1/{AATAT}$_n$ satellite in *Drosophila*. Interestingly, we found that *Drosophila* D1 protein localizes to major satellite/chromocenters when ectopically expressed in multiple mouse cell lines (*Figure 1G*, *Figure 1—figure supplement 1C,D*), suggesting that D1 and HMGA1 may possess an orthologous and conserved function as satellite DNA/chromocenter-binding proteins.

## D1 and HMGA1 are required for organizing chromocenters

We next examined the effects of *D1* mutation and siRNA-mediated knockdown of HMGA1 on chromocenters. We used two *D1* alleles, $D1^{LL03310}$ and $D1^{EY05004}$, which we show to be protein null alleles, evidenced by near-complete loss of anti-D1 antibody staining (*Figure 1—figure supplement 2A–C*). When these alleles were combined with the D1 deficiency allele, Df(3R)BSC666, it led to severe declustering of {AATAT}$_n$ satellite DNA (*Figure 1H–J*, *Figure 1—figure supplement 2D–E*), suggesting that D1 is required for clustering of pericentromeric satellite DNA into chromocenters. We observed D1's requirement for chromocenter formation in multiple cell types (*Figure 1—figure supplement 2F–I*), but we largely focused on spermatogonial cells, where the phenotypes (such as cell death) were most penetrant and severe.

We also examined the requirement for HMGA1 in mouse chromocenter formation. Following siRNA-mediated knockdown of HMGA1, which led to near complete loss of HMGA1 protein (see *Figure 2D,E* and *Figure 2—figure supplement 1A–B,D–E* for efficiencies of HMGA1 knockdown), we observed chromocenter disruption in multiple mouse cell lines (*Figure 1K–M*, *Figure 1—figure supplement 2J–L*). These results suggest that D1 and HMGA1 have an orthologous function to organize pericentromeric satellite DNA into chromocenters.

## Loss of D1/HMGA1 leads to micronuclei formation

To explore the function of chromocenters and satellite DNA, we examined the effects of *D1* mutation/HMGA1 knockdown, which showed strikingly similar phenotypes. We found that *D1* mutation as well as siRNA-mediated HMGA1 knockdown in multiple mouse cell lines resulted in a significant increase in micronuclei formation (*Figure 2A–F*, *Figure 2—figure supplement 1A–F*).

Micronuclei are known to have compromised nuclear envelope integrity, leading to DNA damage and catastrophic chromosomal rearrangement therein (*Crasta et al., 2012*; *Hatch et al., 2013*). Therefore, we first examined a possible defect in nuclear envelope integrity in *D1* mutant. We found that loss of D1 led to breaching of the nuclear envelope both in major and micronuclei, visualized by the cytoplasmic leakage of nuclear GFP (nlsGFP) (*Figure 2G–I*), suggesting that nuclear envelope integrity might be generally compromised. Consistently, we observed mislocalization of nuclear envelope proteins in *D1* mutant spermatogonia. We frequently observed that lamin surrounded the nucleus incompletely in *D1* mutant (1.9% in control (n = 52) and 68.9% in *D1* mutant (n = 58)) (*Figure 2J,K*, arrows indicate lamin-negative regions on the nuclear membrane). We also observed cytoplasmic 'holes', which resemble the nucleus in that they exclude cytoplasmic proteins such as

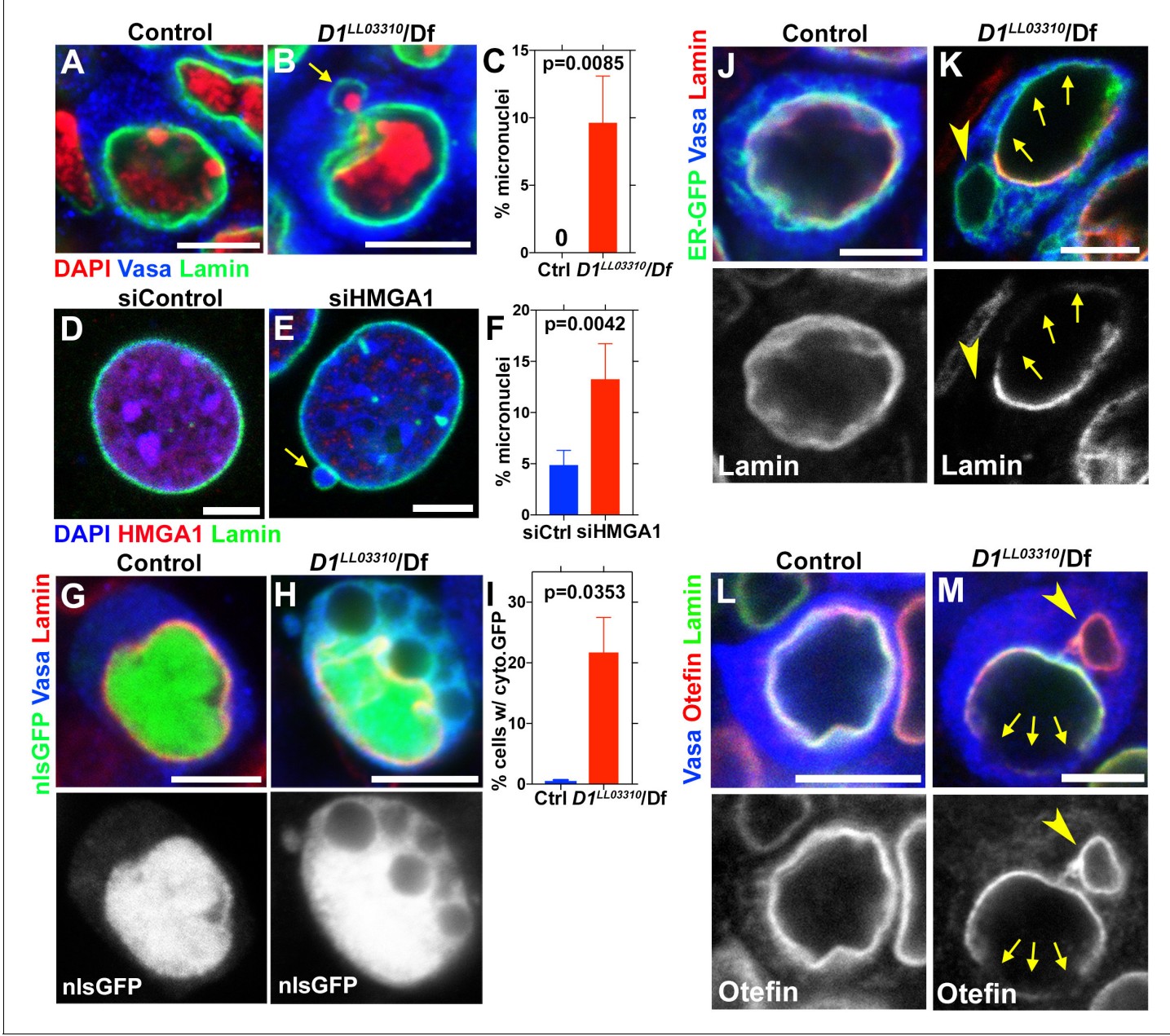

**Figure 2.** D1/HMGA1 loss-of-function results in micronuclei formation, and defective nuclear envelope integrity. (A, B) Control ($D1^{LL03310}$/+) (A) and $D1^{LL03310}$/Df mutant (B) spermatogonial cells stained for DAPI (red), Vasa (blue) and LaminDm$_0$ (green). Arrow indicates micronucleus. Bars: 5 µm. (C) Quantification of micronuclei-containing cells from +/+ control (n = 269) and $D1^{LL03310}$/Df (n = 334) from three independent experiments. p-Value from student's t-test is shown. Error bars: SD. (D, E) siControl (D) and siHMGA1 transfected (E) MOVAS cells stained for DAPI (blue), HMGA1 (red) and Lamin (green). Arrow indicates micronucleus. (F) Quantification of micronuclei-containing cells in siControl (n = 518) and siHMGA1 (n = 588) transfected cells from four independent experiments. (G, H) Control ($D1^{LL03310}$/+) (G) and $D1^{LL03310}$/Df (H) spermatogonia expressing nls-GFP (green) stained for Vasa (blue) and LaminDm$_0$ (red). nlsGFP was observed in cytoplasm in $D1^{LL03310}$/Df spermatogonia. (I) Quantification of spermatogonia with cytoplasmic GFP (>1 µm exclusions or pan-cytoplasmic) in $D1^{LL03310}$/+ (n = 810) and $D1^{LL03310}$/Df (n = 780) testes from two independent experiments. (J, K) $D1^{LL03310}$/+ (J) and $D1^{LL03310}$/Df (K) spermatogonia expressing ER-GFP marker (green) stained for Vasa (blue) and LaminDm$_0$ (red). Arrowhead points to ER marker-positive micronucleus. Arrows point to site of weak nuclear LaminDm$_0$ staining. (L, M) Control ($D1^{LL03310}$/+) (L) and $D1^{LL03310}$/Df (M) spermatogonia stained for Vasa (blue) and LaminDm$_0$ (green) and Otefin (red). Arrowhead points to Otefin-containing micronucleus. Arrows point to site of weak nuclear LaminDm$_0$ staining.

DOI: https://doi.org/10.7554/eLife.34122.006

The following figure supplement is available for figure 2:

**Figure supplement 1.** Formation of micronuclei upon chromocenter disruption in C3H10T1/2 and C2C12 mouse cells.

*Figure 2 continued on next page*

Figure 2 continued

DOI: https://doi.org/10.7554/eLife.34122.007

Vasa (*Figure 2K*, arrowhead), but are devoid of nuclear lamin (*Figure 2K*, arrowhead). These 'holes' were often surrounded by an ER marker, which normally surrounds the nuclear envelope (*Figure 2J*) (*Dorn et al., 2011*). Similarly, Otefin, an inner nuclear membrane LEM-domain protein (*Barton et al., 2014*), also showed perturbed localization (2.7% in control (n = 109) and 24.5% in *D1* mutant (n = 106)) (*Figure 2L,M*, arrows indicate lamin/Otefin negative regions on the nuclear envelope while the arrowhead indicates Otefin-positive micronuclei). Taken together, these results show that *D1* mutant cells exhibit compromised nuclear envelope integrity, which is associated with micronuclei formation.

## Loss of D1/HMGA1 leads to accumulation of DNA damage

It has been shown that defects in nuclear envelope integrity can lead to extensive DNA damage in the major nucleus and micronuclei (*Crasta et al., 2012*; *Hatch et al., 2013*; *Zhang et al., 2015*; *Denais et al., 2016*; *Raab et al., 2016*). Nuclear envelope defects and extensive DNA damages therein lead to catastrophic chromosomal breaks/rearrangements termed chromothripsis (*Crasta et al., 2012*; *Hatch et al., 2013*). Such catastrophic DNA breaks/rearrangements are speculated to lead to tumorigenesis (*Hatch and Hetzer, 2015*).

Consistent with defective nuclear envelope integrity, we observed extensive DNA damage (revealed by γ-H2Av) in both major and micronuclei (*Figure 3A–F*, arrows point to damaged DNA in micronuclei in B and D). Likely as a result of DNA damage and defective nuclear envelope integrity, *D1* mutant testes rapidly degenerated (*Figure 3—figure supplement 1A,B*). When *Omi*, a gene required to promote germ cell death (*Yacobi-Sharon et al., 2013*), was knocked down in *D1* mutant testes, it restored the cellularity in *D1* mutant testis (*Figure 3—figure supplement 1C–D*), but the surviving cells showed a dramatic increase in DNA damage (*Figure 3—figure supplement 1E–F*). Under these conditions, we observed that surviving germ cells in *D1* mutant testes showed a high frequency of chromosome breaks compared to control, revealed by FISH on metaphase chromosome spreads from spermatocytes (3.7% in control (n = 27) vs. 15.8% in *D1* mutant (n = 57)) (*Figure 3G,H*, arrowheads indicate sites of chromosome breaks). These results show that loss of D1/HMGA1 results in compromised nuclear envelope integrity, leading to extensive DNA damage and chromosomal breaks.

## Micronuclei formation in *D1* mutant/HMGA1 knockdown cells is due to budding from the nucleus during interphase

It has been shown that micronuclei form by lagging chromosomes (*Crasta et al., 2012*). Thus, we examined whether *D1* mutation/HMGA1 knockdown resulted in mitotic chromosome segregation errors, causing micronuclei formation. However, we did not observe an increase in lagging chromosomes in *D1* mutant spermatogonia or HMGA1-depleted mouse cells (*Figure 4—figure supplement 1A–G*), suggesting an alternative route for micronuclei formation. Instead, time-lapse live observation showed that micronuclei formed by budding from the interphase nucleus both in *Drosophila* spermatogonia and mouse cells (*Figure 4A–D*). In *Drosophila* spematogonia, nuclear contents were visualized by a GFP-tagged nuclear protein, Df31, and RFP-tagged histone H2Av. Control cells stably maintained nuclear contents for a prolonged time period (only 1 event of nuclear blebbing without concurrent micronuclei formation (as detected by H2Av-RFP) over 1552 min of live imaging) (*Figure 4A*). In contrast, *D1* mutant cells showed budding off of nuclear contents and micronuclei formation in interphase (15 nuclear breaches with eight micronuclei formed over 3427 min of live imaging with a total budding duration of 172 min) (*Figure 4B*). Similarly, live imaging in mouse cells using the Hoechst DNA dye revealed that HMGA1 knockdown also resulted in micronuclei formation during interphase (siControl – no micronuclei formation over 253 min of live observation, siHMGA1 – three micronuclei formed by budding over 5962 min of live imaging with a total budding duration of 310 min) (*Figure 4C,D*). These results show that micronuclei in *D1* mutant/HMGA1-knockdown cells are generated during interphase, via budding from the nucleus.

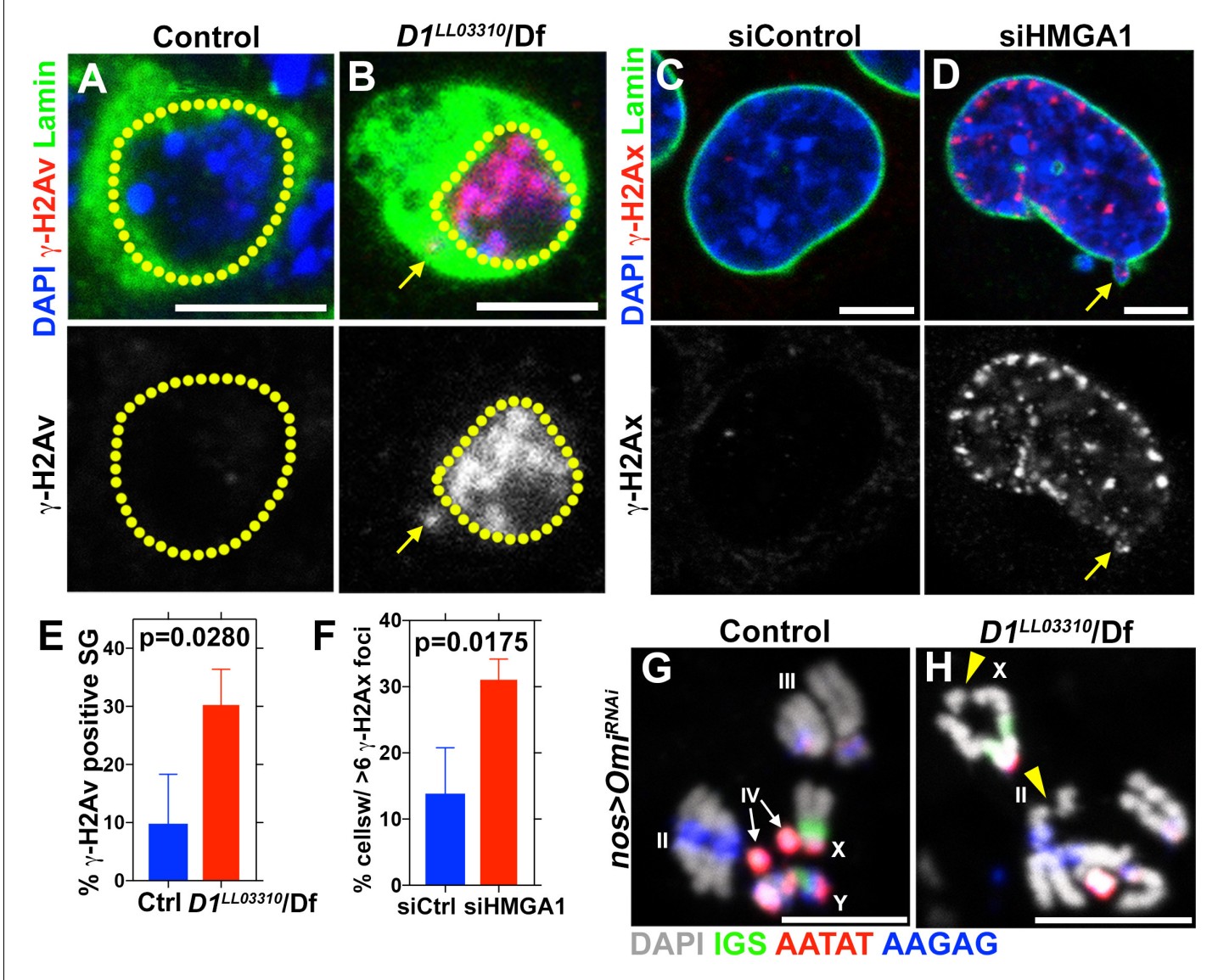

**Figure 3.** D1 mutation/HMGA1 depletion leads to an increase in DNA damage. (**A, B**) Control (*D1^{LL03310}*/+) (**A**) and *D1^{LL03310}*/Df (**B**) spermatogonia stained for DAPI (blue), Vasa (green) and γ-H2Av (red). Dotted lines indicate nucleus and arrow points to DNA damage in micronuclei. (**C, D**) siControl (**C**) and siHMGA1 (**D**) transfected MOVAS cells stained for DAPI (blue), γ-H2Av (red) and LaminDm$_0$ (green). Arrow points to DNA damage in micronuclei. (**E**) Quantification of γ-H2Av positive cells in *D1^{LL03310}*/+ (n = 317) and *D1^{LL03310}*/Df (n = 242) spermatogonia from three independent experiments. (**F**) Quantification of cells containing >6 γ-H2Ax foci in siControl (n = 304) and siHMGA1 (n = 309) transfected cells from three independent experiments. (**G, H**) FISH against the rDNA intergenic spacer (IGS) (green), {AATAT}$_n$ (red) and {AAGAG}$_n$ (blue) on chromosome spreads from meiotic spermatocytes from control (*nos > Omi^{RNAi}*, n = 27) and *D1* mutant (*nos >Omi ^{RNAi}; D1^{LL03310}*/Df, n = 57) testes co-stained for DAPI (grey). *Omi^{RNAi}* was used to block DNA damage-induced cell death. Arrowheads point to chromosome breaks.

DOI: https://doi.org/10.7554/eLife.34122.008

The following source data and figure supplement are available for figure 3:

**Source data 1.** Quantification of g-H2Ax foci in mouse cells.
DOI: https://doi.org/10.7554/eLife.34122.010

**Figure supplement 1.** Chromocenter disruption results in germ cell death in *Drosophila* in an Omi-dependent manner.
DOI: https://doi.org/10.7554/eLife.34122.009

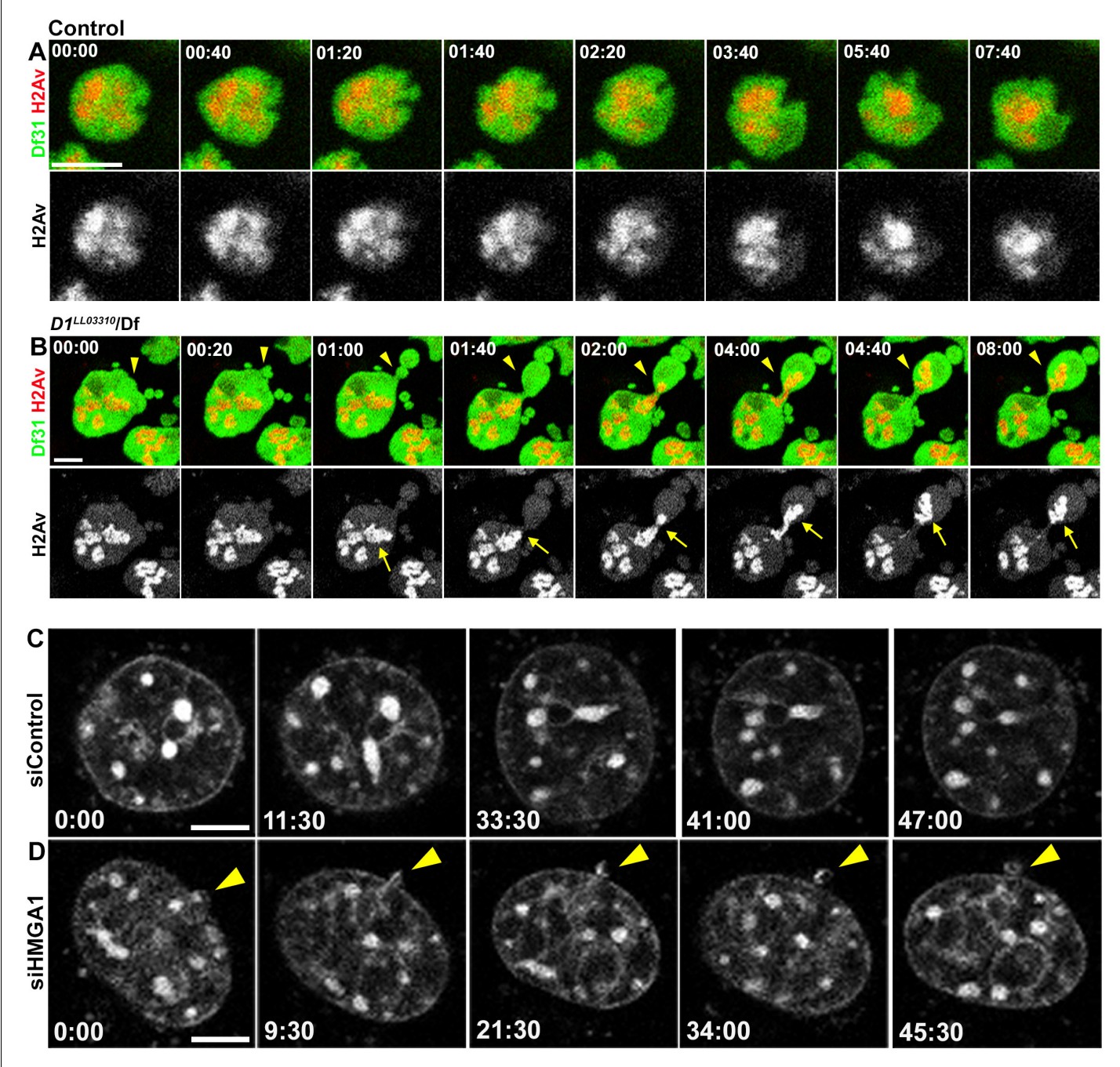

**Figure 4.** D1/HMGA1 loss of function results in micronuclei formation due to nuclear budding during interphase. (A, B) Time-lapse live imaging of control (+/+) (A) and *D1^{LL03310}/Df* (B) spermatogonial cells expressing Df31-GFP as a nuclear marker and H2Av-RFP as a DNA marker. (C, D) Time-lapse live imaging of siControl (C) and siHMGA1 (D) MOVAS cells stained with Hoechst 33342. Arrowheads indicate site of micronucleus budding. Time is indicated in mm:ss. Scale bars: 5 μm.

DOI: https://doi.org/10.7554/eLife.34122.011

The following figure supplement is available for figure 4:

**Figure supplement 1.** Micronuclei formation upon chromocenter disruption is not a result of mitotic lagging chromosomes.
DOI: https://doi.org/10.7554/eLife.34122.012

# D1 bundles satellite DNA from multiple chromosomes to form chromocenter

Based on the above results, we postulated that chromocenter formation, that is clustering of satellite DNA from multiple chromosomes, might be a mechanism to bundle heterologous chromosomes together to prevent individual chromosomes from floating out of the nucleus. In this manner, the full set of chromosomes may be retained within a single nucleus. In the absence of chromocenter formation, individual chromosomes may bud off the nucleus, leading to micronuclei formation.

Previous in vitro experiments indicated that HMGA1 is capable of crosslinking multiple DNA strands with individual AT-hooks binding AT-rich DNA strands (*Vogel et al., 2011*). Bundling of DNA in this manner by D1/HMGA1 could explain how pericentromeric satellite DNA from multiple chromosomes may be clustered to form chromocenters. A few lines of evidence support this idea. When *Drosophila* D1 was expressed in mouse cells, it localized to the chromocenter as described above (*Figure 1G*), and its overexpression enhanced chromocenter formation in a dose-dependent

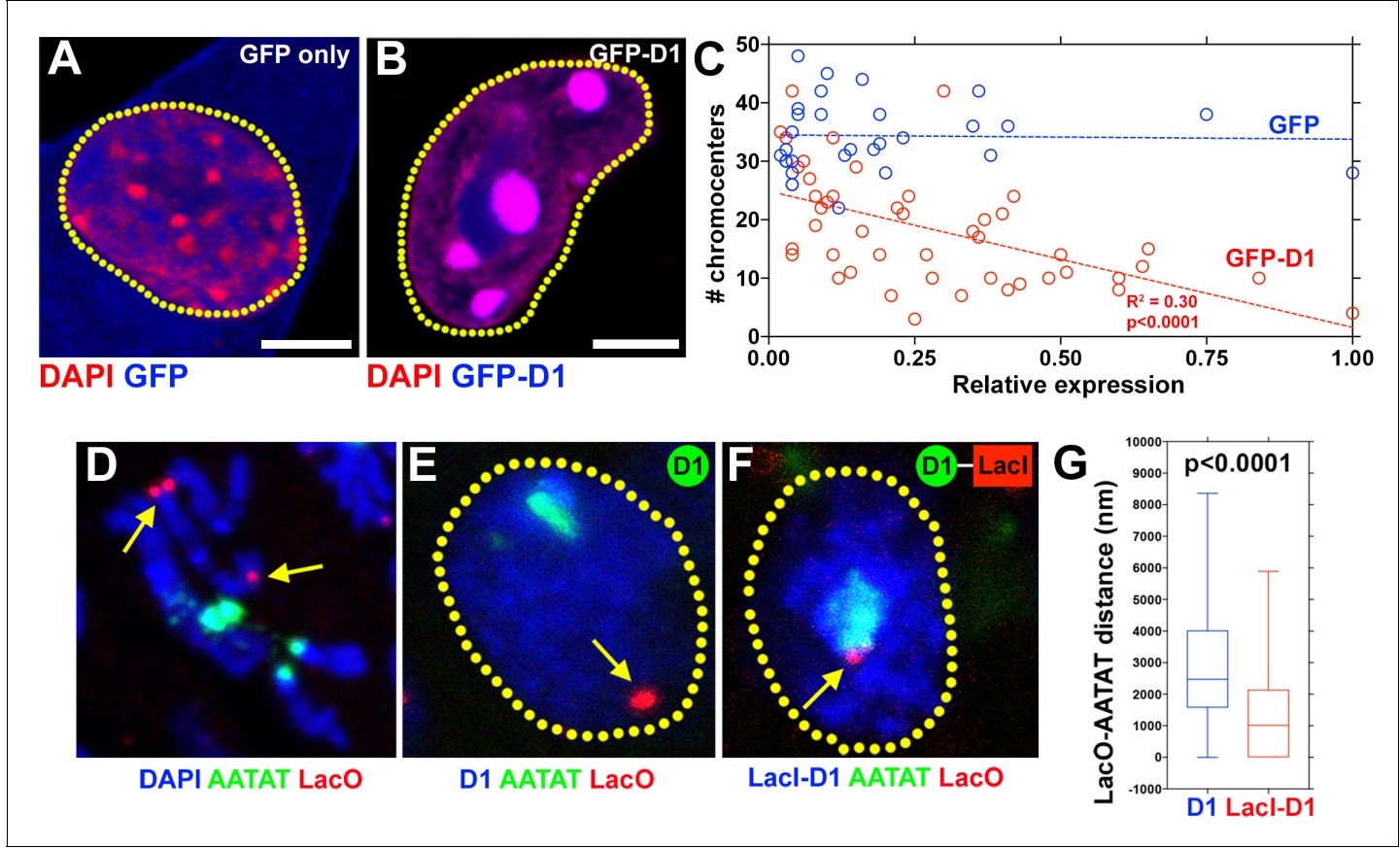

**Figure 5.** D1 bundles satellite DNA from heterologous chromosomes to form chromocenter. (A, B) C2C12 cells expressing GFP only (blue) (A) or GFP-D1 (blue) (B) stained for DAPI (red). Dotted lines indicate nucleus. (C) Quantification of chromocenter number relative to expression level of GFP (n = 29) or GFP-D1 (n = 47). P value and $R^2$ value are indicated from linear regression analysis. (D) FISH against LacO (red) and {AATAT}$_n$ (green) on mitotic neuroblast chromosomes from the LacO strain stained for DAPI (blue), indicating the sites of LacO insertion (arrows). (E, F) FISH against LacO (red) and {AATAT}$_n$ (green) in spermatogonia expressing GFP-D1 (blue) (E) or GFP-LacI-D1 (blue) (F). Arrows indicate location of LacO sequence. (G) AATAT-LacO distance (nm) in GFP-D1 (n = 97) and GFP-LacI-D1 (n = 69) expressing spermatogonia. P value from student's t-test is shown. Error bars: SD. All scale bars: 5 μm.

DOI: https://doi.org/10.7554/eLife.34122.013

The following source data is available for figure 5:

**Source data 1.** Quantification of the relative expression of GFP/GFP-D1 and number of chromocenters in mouse cells.
DOI: https://doi.org/10.7554/eLife.34122.014

**Source data 2.** Quantification of LacO-AATAT distance (nm) in cells expressing GFP-D1 and GFP-LacI-D1.
DOI: https://doi.org/10.7554/eLife.34122.015

manner (*Figure 5A–C*): the higher the amount of D1 that was expressed in mouse cells, the fewer chromocenters per cell was observed (i.e. more clustering). These results suggest that D1 is sufficient to bundle its binding target, tethering it to chromocenter. Consistent with this idea, we found that artificial tethering of D1 protein to euchromatic LacO repeat DNA sequences was sufficient to bring LacO repeats to the chromocenter. D1 protein or D1-LacI fusion protein was expressed in a *Drosophila* strain in which LacO repeats are inserted in the distal regions of the 2nd chromosome (*Figure 5D*, arrows). In control spermatogonial cells expressing wild type D1, LacO repeats were observed far away from the $\{AATAT\}_n$ satellite foci/chromocenters (*Figure 5E,G*, arrow indicates site of LacO repeats in interphase nucleus). However, in cells expressing the LacI-D1 chimeric protein, we observed recruitment of the LacO repeats close to $\{AATAT\}_n$/chromocenters (*Figure 5F,G*, arrow indicates site of LacO repeats recruited to the chromocenter), demonstrating that D1's binding to a DNA sequence is sufficient to incorporate the target sequence into chromocenters.

Although it cannot be visualized how DNA strands from multiple chromosomes might be bundled in these interphase chromocenters, deconvolution microscopy of D1/HMGA1 proteins on early mitotic chromosomes revealed proteinaceous threads between chromatin in the process of condensation (*Figure 6A,B*, arrows indicate D1/HMGA1 threads), which we speculate contributed to bundling of chromosomes in the previous interphase. These threads were also detectable by DNA FISH against $\{AATAT\}_n$ and the mouse major satellite (*Figure 6C,D*, dotted lines are alongside the satellite DNA threads), suggesting that satellite DNA bound by D1/HMGA1 can form threads. These threads likely connect heterologous chromosomes, as we see threads between chromosomes that are clearly distinct in their morphology (e.g. *Figure 6C*). These D1/HMGA1 threads are reminiscent of 'DNA fibers', which were observed among mitotic chromosomes, although their function has never been appreciated (*Takayama, 1975*; *Burdick, 1976*; *Kuznetsova et al., 2007*).

Taken together, these results support a model, in which D1/HMGA1 bind their target sequences (satellite DNA) on multiple chromosomes and bundle them into chromocenters, likely via their multivalent DNA-binding domains (multiple AT-hooks) (*Figure 6E*).

## Discussion

The function of chromocenters, as well as that of satellite DNA, has remained enigmatic, even though cytological association of pericentromeric satellite DNA into chromocenters was identified almost 50 years ago (*Jones, 1970*; *Pardue and Gall, 1970*). Pericentromeric heterochromatin has most often been studied and discussed in the context of how to maintain its heterochromatic, repressed nature (*Nishibuchi and Déjardin, 2017*), based on the assumption that the underlying sequences are mostly selfish, which have negative phenotypic consequences when derepressed in cells (*Zeller and Gasser, 2017*).

Although satellite DNA's function has been speculated and implicated in several examples (*Yunis and Yasmineh, 1971*; *Bonaccorsi et al., 1990*; *Dernburg et al., 1996*; *Menon et al., 2014*), the non-coding nature and lack of conservation in repeat sequence among closely related species led to the idea that they are mostly junk DNA, serving no essential function (*Walker, 1971*; *Doolittle and Sapienza, 1980*). Instead, we propose that satellite DNA is a critical constituent of eukaryotic chromosomes to ensure encapsulation of all chromosomes in interphase nucleus. Our results may also explain why the sequences of pericentromeric satellite DNA are so divergent among closely related species, a contributing factor that led to their dismissal as junk. Based on our model that pericentromeric satellite DNA serves as a platform for generating heterologous chromosome association to form chromocenters, the essential feature of satellite DNA is that they are bound by protein(s) capable of bundling multiple DNA strands. If so, the underlying sequence does not have to be conserved. Instead, the binding of satellite DNA by a chromocenter bundling protein may be a critical feature of pericentromeric satellite DNAs. Based on this idea, chromocenter bundling proteins and pericentromeric satellite DNA may be co-evolving.

We observed perturbation of nuclear envelope integrity upon chromocenter disruption. Understanding the mechanisms underlying perturbation of nuclear envelope integrity in *D1* mutant awaits future investigation. Previous studies have documented that cytoskeletal forces are transmitted to chromatin through nuclear envelope and external mechanical forces can cause temporary nuclear envelope breaches (*King et al., 2008*; *Denais et al., 2016*; *Hatch and Hetzer, 2016*; *Raab et al., 2016*). Therefore, we speculate that chromosome bundling in the form of chromocenter may help

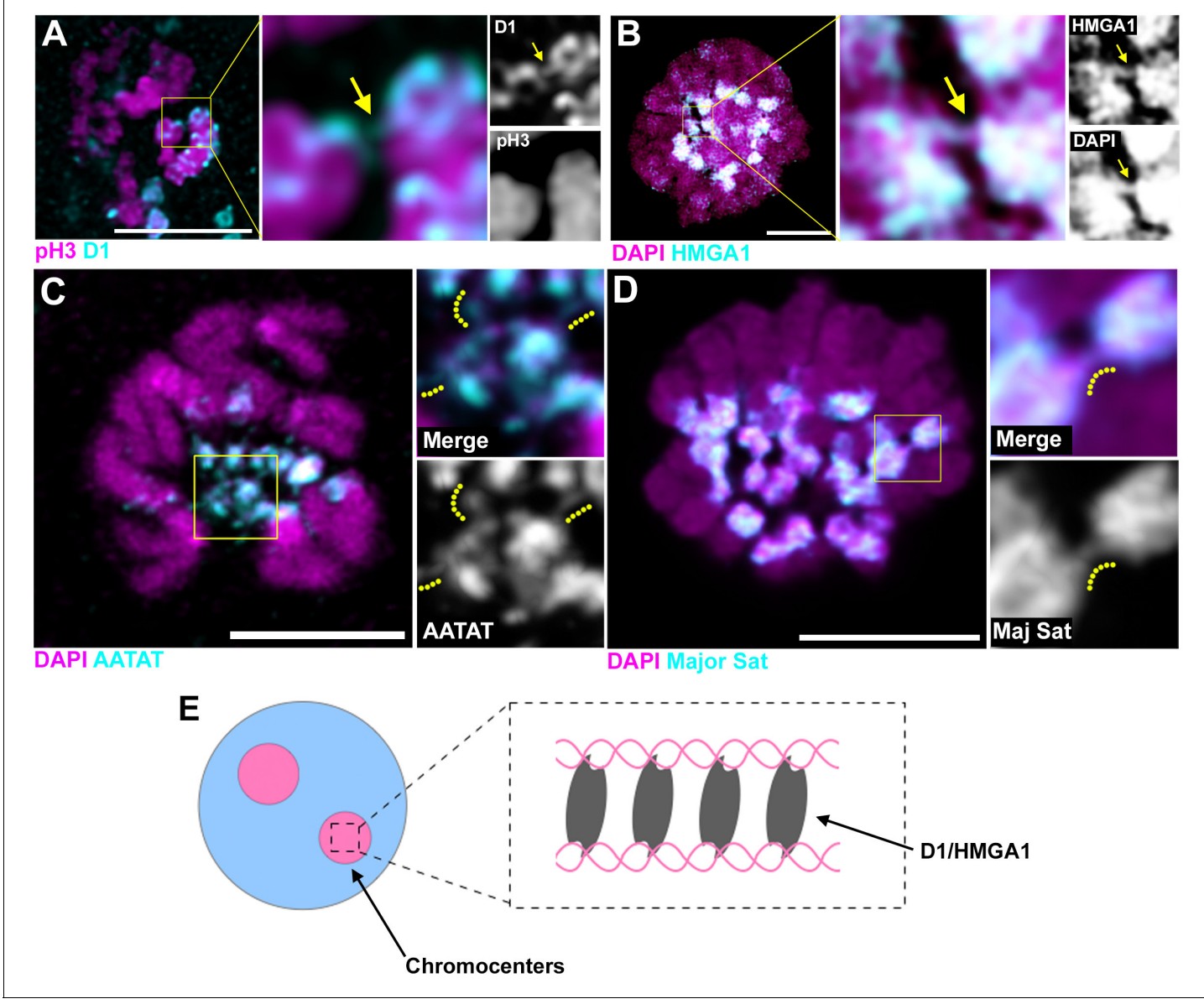

**Figure 6.** D1/HMGA1 and satellite DNA form chromatin threads that link chromosomes. (**A**) Deconvolution microscopy performed on *Drosophila* mitotic neuroblasts stained for D1 (cyan) and pH3-S10 (magenta). Arrows in magnified images indicate D1-positive thread connecting two chromosomes. (**B**) Deconvolution microscopy performed on CSK-extracted RAW 264.7 macrophages stained for HMGA1 (cyan) and DAPI (magenta). Arrows in magnified images indicate HMGA1-positive thread connecting two chromosomes. (**C**) Deconvolution microscopy performed on neuroblast mitotic chromosomes stained for DAPI (magenta) and FISH against $\{AATAT\}_n$ (cyan) from a *Drosophila* strain containing AATAT-rich B chromosomes (***Bauerly et al., 2014***). Dotted lines in magnified images indicate AATAT-positive threads connecting heterologous chromosomes. (**D**) Deconvolution microscopy performed on RAW 264.7 macrophages stained for DAPI (magenta) and FISH against major satellite (cyan). Dotted lines in magnified images indicate major satellite-positive threads connecting two chromosomes. (**E**) The model of chromosome bundling by D1/HMGA1 and satellite DNA.

DOI: https://doi.org/10.7554/eLife.34122.016

prevent cytoskeletal forces from shearing chromosomes and nuclear envelope: when chromosomes are not bundled, cytoskeletal forces may be transmitted to individual chromosomes and associated nuclear envelope, resulting in shearing of nuclear envelope, disrupting its integrity.

In summary, our study provides the first evidence for a conserved function of pericentromeric satellite DNA and chromocenters. Our data suggest that the multi-AT-hook proteins, D1 and HMGA1, play an evolutionarily conserved role in the formation of chromocenters, likely via their ability to bind

and bundle satellite DNA from heterologous chromosomes. Heterologous chromosome association, mediated by chromocenter-binding proteins, may represent a third mode of chromosomal 'gluing' after meiotic homologous pairing and sister chromatid cohesion. Through heterologous association, the chromocenter plays a fundamental role in maintaining the full complement of the genome, which is divided into multiple chromosomes, into a single nucleus. This function of the chromocenter may be conserved in eukaryotic species that contain pericentromeric satellite DNA, thereby bringing about a signature characteristic of eukaryotic cells.

# Materials and methods

**Key resources table**

| Reagent type (species) or resource | Designation | Source or reference | Identifiers | Additional information |
|---|---|---|---|---|
| Genetic Reagent (*D. melanogaster*) | D1$^{EY05004}$ | Bloomington Drosophila Stock Center | ID_BDSC:17340 | |
| Genetic Reagent (*D. melanogaster*) | Df(3R)BSC666 | Bloomington Drosophila Stock Center | ID_BDSC:26518 | |
| Genetic Reagent (*D. melanogaster*) | UAS-Omi$^{RNAi}$ | Bloomington Drosophila Stock Center | ID_BDSC:55165 | |
| Genetic Reagent (*D. melanogaster*) | UAS-GFP-nls | Bloomington Drosophila Stock Center | ID_BDSC:4776 | |
| Genetic Reagent (*D. melanogaster*) | UAS-GFP-ER-SR | Bloomington Drosophila Stock Center | ID_BDSC:59042 | |
| Genetic Reagent (*D. melanogaster*) | D1$^{LL03310}$ | Kyoto Stock Center | ID_DGRC:140754 | |
| Genetic Reagent (*D. melanogaster*) | Df31-GFP | Kyoto Stock Center | ID_DGRC:110806 | |
| Genetic Reagent (*D. melanogaster*) | nos-gal4 | PMID: 9501989 | | |
| Genetic Reagent (*D. melanogaster*) | hs-flp;nos-FRT-stop-FRT-gal4,UAS-GFP | PMID: 24465278 | | |
| Genetic Reagent (*D. melanogaster*) | UAS-H2A-YFP | PMID: 11146626 | | |
| Genetic Reagent (*D. melanogaster*) | B1 LacO | PMID: 12225662 | | |
| Genetic Reagent (*D. melanogaster*) | mtrm$^{126}$ + B | PMID: 24478336 | | Gift of Dr. Scott Hawley |
| Recombinant DNA Reagent | pUASt-GFP-attB | PMID: 24465278 | | |
| Recombinant DNA Reagent | pUASt-GFP-D1-attB | This Paper | | |
| Recombinant DNA Reagent | pUASt-GFP-LacI-D1-attB | This Paper | | |
| Recombinant DNA Reagent | pCDNA3 | | | Gift of Dr. Cheng-Yu Lee |
| Cell Line | MOVAS | | | Gift of Dr. Daniel Eitzman |
| Cell Line | C2C12 | | | Gift of Dr. David Bridges |
| Cell Line | RAW264.7 | | | Gift of Dr. Harry Mobley |
| Cell Line | C3H10T1/2 | | | Gift of Dr. Stephen Weiss |
| siRNA | ON-TARGET plus Mouse HMGA1 siRNA SMARTpool | Dharmacon/GE Healthcare | ID_Dharmacon: L-049293–01 | |
| siRNA | ON-TARGET plus Non-targeting pool | Dharmacon/GE Healthcare | ID_Dharmacon: L-001810–10 | |

*Continued on next page*

*Continued*

| Reagent type (species ) or resource | Designation | Source or reference | Identifiers | Additional information |
|---|---|---|---|---|
| Antibody | anti-Vasa | Santa Cruz Biotechnology | ID_SCB: d-26 | |
| Antibody | anti-H3K9 dimethyl | Abcam | ID_abcam: ab32521 | |
| Antibody | anti-Otefin | | | Gift of Dr. Georg Krohne |
| Antibody | anti-D1 | This Paper | | Peptide - CDGENDAND GYVSDNYNDSESVAA |
| Antibody | anti-LaminDm$_0$ | Developmental Studies Hybridoma Bank | ID_DSHB: ADL84.12 | |
| Antibody | anti-$\gamma$-H2Av | Developmental Studies Hybridoma Bank | ID_DSHB: UNC93-5.2.1 | |
| Antibody | Phalloidin-Alexa546 | ThermoFisher | ID_Thermo Fisher: a22283 | |
| Antibody | anti-HMGA1 | Abcam | ID_abcam: ab129153 | |
| Antibody | anti-LaminB (C20) | Santa Cruz Biotechnology | ID_SCB: 2616 | |
| Antibody | anti-$\alpha$-tubulin | Developmental Studies Hybridoma Bank | ID_DSHB: 4.3 | |
| Antibody | anti-$\gamma$-H2Ax S139 | Cell Signaling Technologies | ID_CST: 2577 | |

## Fly husbandry and strains

All fly stocks were raised on standard Bloomington medium at 25°C. The following fly stocks were used: $D1^{EY05004}$(BDSC17340), Df(3R)BSC666 (BDSC26518), *UAS-Omi$^{RNAi}$* (BDSC55165), *UAS-GFP-nls* (BDSC4776) and *UAS-GFP-ER-SR* (BDSC59042) were obtained from the Bloomington *Drosophila* stock center. $D1^{LL03310}$ (DGRC140754) and Df31-GFP (DGRC110806) were obtained from the Kyoto stock center. *nos-gal4* (**Van Doren et al., 1998**), *hs-flp;nos-FRT-stop-FRT-gal4,UAS-GFP* (**Salzmann et al., 2013**), *UAS-H2A-YFP* (**Bellaïche et al., 2001**) and *B1 LacO* (**Vazquez et al., 2002**) have been previously described. A stock containing B chromosomes, *mtrm$^{126}$ +B* (**Bauerly et al., 2014**), was a kind gift from Scott Hawley. Chromocenter disruption was scored in *Drosophila* testes by assessing {AATAT}$_n$ morphology in GFP +cells that were generated as follows in control (hs-flp; *nos-FRT-stop-FRT-gal4,UAS-GFP*) and D1 mutant (*hs-flp;nos-FRT-stop-FRT-gal4, UAS-GFP; D1$^{LL03310}$/Df*) flies. Testes were dissected 24 hr following a 20 min heat shock at 37°C. Chromocenters were considered disrupted in *Drosophila* and mouse when satellite DNA adopted thread-like morphology in interphase nuclei. Micronuclei were scored in 0-3 d testes where early germ cell chromosomes were labeled with H2A-YFP. The genotypes used were, control – *nos > H2 A-YFP* and D1 mutant – *nos > H2 A-YFP; D1LL03310/Df*.

## Transgene construction

For construction of *UAS-GFP-D1*, the *D1* ORF was PCR-amplified from cDNA using the following primer pair, 5'-GATCAGATCTATGGAGGAAGTTGCGGTAAAG-3' and 5'-GATCCTCGAGTTAGG-CAGCTACCGATTCGG-3'. The amplified fragment was subcloned into the BglII and XhoI sites of *pUASt-EGFP-attB* (**Salzmann et al., 2013**) resulting in *UAS-GFP-D1*. For *UAS-GFP-LacI-D1*, the *LacI* ORF (lacking 11 C-terminal residues) (**Straight et al., 1996**) was synthesized using GeneArt (Thermofisher) and inserted into the BglII site of *UAS-GFP-D1* resulting in *UAS-GFP-LacI-D1*. Transgenic flies were generated by PhiC31 integrase-mediated transgenesis into the *attP40* site (BestGene). For expression of GFP and GFP-D1 in mouse cells, *GFP* and *GFP-D1* was subcloned from *pUASt-EGFP-attB* into pCDNA3 (gift from Cheng-Yu Lee) using EcoRI and XhoI sites.

## Cell lines

Mouse MOVAS cells were obtained from Daniel Eitzman. Mouse C2C12 cells were obtained from David Bridges. Mouse RAW264.7 cells were obtained from Dr. Harry Mobley. Mouse C3H10T1/2 cells were obtained from Stephen Weiss. MOVAS, C2C12 and RAW264.7 cells were maintained in

Dulbecco's minimal essential medium (DMEM) (Gibco) supplemented with 10% fetal bovine serum (FBS). C3H10T1/2 cell line was maintained in alpha minimal essential media (Gibco) supplemented with 10% fetal bovine serum. All cell lines used were authenticated as mouse cells by the presence of mouse-specific satellite DNA as is shown throughout the manuscript. Two major cell lines used in this study, C2C12 and MOVAS cells, were treated with Plasmocin (Invivogen) prior to use as a precaution for mycoplasma infection.

## siRNA and transfections

RNA interference (RNAi) against HMGA1 was performed using ON-TARGET plus Mouse HMGA1 siRNA SMARTpool (Dharmacon, L-049293–01) consisting of the following target sequences, CCA UUUAGCCGCAGCCCGA, AGGCAAACGGGCACCAACA, GGGCGCAGCAGACUGGUUA, GUUCA UUCUUAGAUACCCA. ON-TARGET plus Non-targeting pool (Dharmacon, D-001810–10) consisting of the following sequences, UGGUUUACAUGUCGACUAA, UGGUUUACAUGUUGUGUGA, UGG UUUACAUGUUUUCUGA, UGGUUUACAUGUUUUCCUA, was used as a negative control. siRNA transfections were performed using DharmaFECT four reagent (Dharmacon, Lafayette, CO) according to the manufacturer's protocol. 25 nM of siControl and siHMGA1 were transfected using DharmaFECT 4 (Dharmacon) according to the manufacturer's protocol. Cells were fixed for immunostaining/in situ hybridization 6 days post-transfection. Transient transfection of GFP and GFP-D1 was performed using Fugene HD (Roche) reagent according to the manufacturer's protocol.

## Immunofluorescence staining and microscopy

For *Drosophila* tissues, immunofluorescence staining was performed as described previously (*Cheng et al., 2008*). Briefly, tissues were dissected in PBS, transferred to 4% formaldehyde in PBS and fixed for 30 min. Testes were then washed in PBS-T (PBS containing 0.1% Triton-X) for at least 60 min, followed by incubation with primary antibody in 3% bovine serum albumin (BSA) in PBS-T at 4°C overnight. Samples were washed for 60 min (three 20 min washes) in PBS-T, incubated with secondary antibody in 3% BSA in PBS-T at 4°C overnight, washed as above, and mounted in VECTA-SHIELD with DAPI (Vector Labs). The following primary antibodies were used: rabbit anti-vasa (1:200; d-26; Santa Cruz Biotechnology), rabbit anti-H3K9 dimethyl (1:200; Abcam, ab32521), guinea pig anti-Otefin (gift from Georg Krohne, 1:400), chicken anti-Cid (1:500, generated using the synthetic peptide CDGENDANDGYVSDNYNDSESVAA (Covance)), mouse anti-LaminDm$_0$ (ADL84.12, 1:200, Developmental Studies Hybridoma Bank), mouse anti-$\gamma$−H2Av (UNC93-5.2.1, 1:400, Developmental Studies Hybridoma Bank), Phalloidin-Alexa546 (ThermoFisher, a22283, 1:200). Adherent mouse cells were fixed in 4% formaldehyde in PBS for 20 min at room temperature on coverslips. Cells were permeabilized in PBS-T for 5 min, rinsed three times with PBS, blocked using 3% BSA in PBS-T for 30 min at room temperature and incubated with primary antibody diluted in 3% BSA in PBS-T overnight at 4°C. Cells were then washed for 30 min (three 10 min washes), incubated with secondary antibody in 3% BSA in PBS-T for 2 hr at room temperature, washed as above and mounted in VECTASHIELD with DAPI (Vector Labs). For nucleoplasmic extraction, cells were incubated with CSK buffer (10 mM PIPES pH7, 100 mM NaCl, 300 mM sucrose, 3 mM MgCl$_2$, 0.5% Triton X-100, 1 mM PMSF) for 10 min at room temperature. After CSK extraction, cells were washed with PBS and fixed and immunostained as above. The following antibodies were used: rabbit anti-HMGA1 (1:400, Abcam, ab129153), goat anti-LaminB (C-20) (1:20, Santa Cruz Biotechnology, sc-2616), mouse anti-$\alpha$-tubulin (4.3, 1:100, Developmental Studies Hybridoma Bank) and $\gamma$−H2Ax S139 (2577, 1:200, Cell Signaling Technologies). Images were taken using a Leica TCS SP8 confocal microscope with 63x oil-immersion objectives (NA = 1.4). Deconvolution was performed when indicated using the Hyvolution package from Leica. Images were processed using Adobe Photoshop software.

## Time-lapse live imaging

Testes from newly eclosed flies were dissected into Schneider's *Drosophila* medium containing 10% fetal bovine serum. The testis tips were placed inside a sterile glass-bottom chamber and were mounted on a three-axis computer-controlled piezoelectric stage. An inverted Leica TCS SP8 confocal microscope with a 63 × oil immersion objective (NA = 1.4) was used for imaging. For mouse live cell imaging, transfected cells were seeded onto a sterile glass-bottom chamber coated with poly-lysine. Cells were incubated with Hoechst 33342 for 10 min, rinsed with PBS and fresh medium was

added to the chamber. Cells were imaged using a stage-top Tokai-Hit incubator that was mounted on an inverted TCS SP5 confocal microscope with a 63x oil immersion objective (NA = 1.4). All images were processed using Adobe Photoshop software. Metrics used for quantification of live imaging were total imaging duration (defined as number of cells x imaging duration), total budding duration (defined as number of cells with micronuclei that formed by budding x time with budded micronuclei).

## DNA fluorescence in situ hybridization

Whole mount *Drosophila* testes were prepared as described above, and optional immunofluorescence staining protocol was carried out first. Subsequently, samples were post-fixed with 4% formaldehyde for 10 min and washed in PBS-T for 30 min. Fixed samples were incubated with 2 mg/ml RNase A solution at 37°C for 10 min, then washed with PBS-T +1 mM EDTA. Samples were washed in 2xSSC-T (2xSSC containing 0.1% Tween-20) with increasing formamide concentrations (20%, 40% and 50%) for 15 min each followed by a final 30 min wash in 50% formamide. Hybridization buffer (50% formamide, 10% dextran sulfate, 2x SSC, 1 mM EDTA, 1 µM probe) was added to washed samples. Samples were denatured at 91°C for 2 min, then incubated overnight at 37°C. For mitotic chromosome spreads, testes and larval 3rd instar brains were squashed according to previously described methods (*Larracuente and Ferree, 2015*). Briefly, tissue was dissected into 0.5% sodium citrate for 5–10 min and fixed in 45% acetic acid/2.2% formaldehyde for 4–5 min. Fixed tissues were firmly squashed with a cover slip and slides were submerged in liquid nitrogen until bubbling ceased. Coverslips were then removed with a razor blade and slides were dehydrated in 100% ethanol for at least 5 min. After drying, hybridization mix (50% formamide, 2x SSC, 10% dextran sulfate, 100 ng of each probe) was applied directly to the slide, samples were heat denatured at 95°C for 5 min and allowed to hybridize overnight at room temperature. Following hybridization, slides were washed thrice for 15 min in 0.2X SSC and mounted with VECTASHIELD with DAPI (Vector Labs). For the in situ experiment described in *Figure 4j–m*, testes were dissected into PBS and fixed in 4% formaldehyde for 4 min. Tips of fixed testes were firmly squashed with a cover slip and slides were submerged in liquid nitrogen until bubbling ceased. Coverslips were removed with a razor blade and slides were subjected to 5 min washes in 2XSSC and 2XSSC with 0.1% Tween-20. The samples were denatured in freshly made 0.07N NaOH for 5 min, rinsed in 2X SSC. Hybridization mix (50% formamide, 2x SSC, 10% dextran sulfate, 100 ng of each probe) was added directly to the slide and allowed to hybridize overnight at room temperature. Following hybridization, slides were washed three times for 15 min in 0.2X SSC and mounted with VECTASHIELD with DAPI (Vector Labs). The following probes were used for *Drosophila* in situ hybridization: {AATAT}$_6$, {AAGAG}$_6$, IGS and have been previously described (*Jagannathan et al., 2017*). LacO probe - 5'-Cy5-CCACAAATTGTTA TCCGCTCACAATTCCAC-3'. For interphase mouse cells, optional immunostaining was carried out as above. Subsequently, samples were post-fixed with 4% formaldehyde in PBS for 10 min and rinsed three times in PBS. Post-fixed cells were incubated with 0.1 mg/ml RNase A solution at 37°C for 1 hr, rinsed three times in PBS and denatured in 1.9M HCl for 30 min. After three rinses in ice-cold PBS, hybridization mix (2X SSC, 60% formamide, 5 µg/ml salmon sperm DNA and 500 nM probe) was added to the samples and incubated overnight at room temperature. Following hybridization, coverslips were washed three times for 15 min in 2X SSC and mounted with VECTASHIELD with DAPI (Vector Labs). For mouse mitotic cells, chromosomes were spread on slides as previously described. Subsequently, chromosomes were denatured in 70% formamide in 2XSSC for 1.5 min at 70°C, dehydrated in 100% ethanol and hybridization mix (2X SSC, 60% formamide, 5 µg/ml salmon sperm DNA and 500 nM probe) was added directly to the slide and incubated overnight at room temperature. Following hybridization, slides were washed three times for 15 min in 2X SSC and mounted with VECTASHIELD with DAPI (Vector Labs). The following probe was used: Major satellite - 5'-Cy3-GGAAAATTTAGAAATGTCCACTG-3'.

## Acknowledgements

We thank Cheng-Yu Lee, Scott Hawley, Stephen Weiss, Harry Mobley, Dave Bridges, Daniel Eitzman, Georg Krohne, Bloomington *Drosophila* Stock Center, Kyoto Stock Center, Developmental Studies Hybridoma Bank and Michigan Imaging Laboratory for reagents and resources. We thank the Yamashita lab members, Sue Hammoud, Ajit Joglekar, Puck Ohi, Dan Barbash, and Maurizio Gatti for

discussion and comments on the manuscript. This research was supported by the Howard Hughes Medical Institute (YY) and an American Heart Association postdoctoral fellowship (MJ). MJ and YY conceived the project, interpreted the data and wrote the manuscript. All authors contributed to conducting experiments and analyzing data.

## Additional information

### Competing interests
Yukiko M Yamashita: Reviewing editor, *eLife*. The other authors declare that no competing interests exist.

### Funding

| Funder | Author |
|---|---|
| Howard Hughes Medical Institute | Yukiko M Yamashita |
| National Institute of General Medical Sciences | Yukiko M Yamashita |
| American Heart Association | Madhav Jagannathan |

The funders had no role in study design, data collection and interpretation, or the decision to submit the work for publication.

### Author contributions
Madhav Jagannathan, Conceptualization, Formal analysis, Supervision, Funding acquisition, Investigation, Writing—original draft, Writing—review and editing; Ryan Cummings, Formal analysis, Validation, Investigation, Writing—review and editing; Yukiko M Yamashita, Conceptualization, Supervision, Funding acquisition, Writing—original draft, Writing—review and editing

### Author ORCIDs
Madhav Jagannathan (iD) http://orcid.org/0000-0003-3428-6812
Ryan Cummings (iD) http://orcid.org/0000-0003-0540-9174
Yukiko M Yamashita (iD) http://orcid.org/0000-0001-5541-0216

### Decision letter and Author response
Decision letter https://doi.org/10.7554/eLife.34122.019
Author response https://doi.org/10.7554/eLife.34122.020

## Additional files

### Supplementary files
• Transparent reporting form
DOI: https://doi.org/10.7554/eLife.34122.017

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
