## [Decision Letter]

Thank you for submitting your article "A conserved function for pericentromeric satellite DNA" for consideration by *eLife*. Your article has been reviewed by three peer reviewers, and the evaluation has been overseen by K VijayRaghavan as the Reviewing Editor and Senior Editor. The following individual involved in the review of your submission has agreed to reveal her identity: Susan Gerbi.

The reviewers have discussed the reviews with one another and the Reviewing Editor has drafted this decision to help you prepare a revised submission.

Summary:

This is an excellent paper with solid and thorough data leading to a conceptually novel model. It includes scholarly citation of papers from the older literature as well as current papers, drawing together many disparate areas. The authors have studied the proteins D1 in *Drosophila* and its counterpart HMGA1 in mouse which are multi AT-hook satellite DNA binding proteins. First, the authors show that these proteins co-localize with chromocenters in *Drosophila* and mouse. Next, they demonstrate that deletions or knock-down of these proteins disrupt the chromocenters. Moreover, they find that defective chromocenter formation leads to budding off of micronuclei from the nucleus. In addition, loss of D1/HMGA1 leads to accumulation of DNA damage. An elegant experiment with an ectopic LacO and a D1/LacI fusion shows that the D1/LacI fusion protein can recruit the ectopic LacO to the chromocenter. Microscopy reveals that D1/HMGA1 can bundle threads of satellite DNA, consistent with their hypothesis that this helps to hold heterologous chromosomes together in the interphase nucleus. Most of their experiments are carried out in both *Drosophila* and mouse, suggesting the universality of the proposed function for the chromocenter.

Although this is primarily a descriptive study, it suggests a mechanism for retention of chromosomes in the nucleus. Several important questions (not articulated by the authors) could be followed up in future studies based on these foundational observations. For example: in *S. pombe* the homolog of ORC4 is an AT-hook protein, thus raising the question of whether D1/HMGA1 associates with other ORC subunits in satellite DNA. Moreover, since a loss of D1/HMGA1 results in chromosome breakage, it would be interesting to know if this is correlated with a defect in DNA replication. Are the DNA breaks throughout the genome, even though D1/HMGA1 is localized in a few places? If so, how is the genome-wide breakage caused? Could the loss of D1/HMGA1 be causal for chromothripsis (shattering of the genome)? There is no need for the authors to address these and other questions in their paper, but it simply illustrates how the present paper may stimulate future experiments by the field.

Overall, the data are convincing and support the authors' model that satellite DNA at pericentromeres form chromocenters, via satellite DNA-binding proteins, to keep the nucleus intact. This paper makes an important contribution and definitely merits publication if the important suggestions below are speedily addressed.

Essential revisions:

1) Some results are not quantified, such as Figure 2J-M. We would expect some sort of quantification for results that merit inclusion in the main figure set.

2) We did not understand panels G and H in Figure 3. Some additional explanation of how the experiment was designed and how it should be interpreted would be helpful. These panels also lack quantification.

3) The live imaging in Figure 4 should have some quantification, such as how frequently budding events were observed. Also, the experiment could be more informative if it were repeated with markers for chromatin (such as histone H2B, which is commonly used) and nuclear envelope, to reveal more details about how nuclear envelope defects lead to budding and whether chromatin is initially in the bud.

4) In Figure 6 threads between chromosomes are shown. These are said to be between heterologous chromosomes in the Discussion, but we don't see data that clearly distinguish between homologous and heterologous chromosomes. If the authors have such data, they should mention it, and if not they should be clear in the Discussion that heterology is an interpretation. The model schematic (Figure 6E) could be more informative. How do the threads combine parts of different chromosomes? Do they consist of DNA from each chromosome cross-linked by D1/HMGA1 proteins? A little more detail in the schematic would help readers understand the model.

5) What is the connection between chromocenters and nuclear envelope defects? It's not obvious why chromosomes need to be linked to keep the nucleus intact. This idea is crucial for the overall message of the paper, so it merits some discussion even if we don't know the answer yet.

6) The first heading in the Results section states that D1 and HMCA1 bind satellite DNA, but the data do not show direct binding. The heading should probably be limited to what is directly shown by the data. Points supported by data in other papers can be explained elsewhere in the text.

7) Do the authors have any ideas about why the D1 phenotypes are most severe in spermatogonial cells (subsection “D1 and HMGA1 are required for organizing chromocenters”, first paragraph)?

8) Can the authors speculate about why the satellite DNA that forms chromocenters is located near the centromere? For the function of linking different chromosomes together, it's not clear why centromere localization would be important.

9) If chromocenter bundling proteins and pericentromeric satellite DNA are co-evolving (line 237), why does the D1 protein (from fly) also function in mouse cells where the satellite DNA sequences are different? If the fly protein can bind diverse satellite DNAs, then what does the co-evolution mean?

10) What happens in mitosis, when chromocenters on different chromosomes should not be linked to each other? Are the D1/HMGA1 proteins removed in mitosis, or is there some other mechanism to inactivate them?

11) Chromocenters are not always as visible in some cells (e.g., human cell lines) as they are in the mouse and fly cells in this paper. Do the authors think they are less important in these cases, or still important but not as apparent by standard DNA staining for some reason?

12) In Figure 1 and Figure 1—figure supplement 2 the authors present data on% disruption of chromocenters, but it is not clear how disruption is defined. Is there a maximum number of foci that are considered to be not disrupted, or a minimum number of foci that are considered to be disrupted? These numbers would presumably differ between *Drosophila* and mouse. In most of the photos the difference seems clear, but supplement Figure 1—figure supplement 2J-K is a case where having a definition of chromocenter disruption would seem to be helpful.

13) The authors' intriguing hypothesis about the function of chromocenters is welcome, but we think they should be slightly more cautious in ascribing universality to it. This should start in the title, which we think would be better as "A conserved function for chromocenters", since in *Drosophila* they have only shown that the AATAT satellite, not other satellites, participates in chromocenter formation. In the Introduction they say "a rationale for the very existence of pericentromeric satellite DNA is still lacking", but selfish DNA is a rationale that we think is widely accepted for some satellites. The use of a more qualified statement like " a clear rationale for the existence of most satellites is lacking" would be more accurate. On the flip side, budding yeast lacks satellites and chromocenters, yet manages to keep 16 chromosomes together, perhaps through attachment of the chromosomes to the spindle pole body. A little nuance in allowing for exceptions would not diminish the appeal of their hypothesis.

14) The authors say that D1 and HMGA1 "may possess an orthologous and conserved function". Are the proteins orthologous? A little more discussion of AT hook proteins and their phylogenetic distribution in eukaryotes would be appreciated, perhaps with a supplemental cartoon of the proteins and their AT hook domains.

15) In the Discussion, do they imagine that all chromocenters are based on AT hook proteins, or might there be other proteins that bundle multiple DNA strands of other satellites? A little more thought about the scope of applicability of their model might improve the Discussion.

---

## [Author Response]

1) Some results are not quantified, such as Figure 2J-M. We would expect some sort of quantification for results that merit inclusion in the main figure set.

Quantification for both sets of panels was reported in the text. Quantification for Figure 2J-K and Figure 2L-M can be found in the last paragraph of the subsection “Loss of D1/HMGA1 leads to micronuclei formation”.

2) We did not understand panels G and H in Figure 3. Some additional explanation of how the experiment was designed and how it should be interpreted would be helpful. These panels also lack quantification.

We apologize for the lack of clarity in our experimental design for Figure 3G-H. We sought to determine whether the DNA damage observed in the micronuclei and main nuclei of *D1* mutant spermatogonia resulted in chromosomal breaks/shattering and whether these breaks occurred on specific chromosomes or satellite DNAs. To do so, we performed FISH on chromosome spreads from meiotic spermatocytes (the developmental outcome of spermatogonia) in control and *D1* mutant testes. In addition, this experiment was performed in testes that were depleted for Omi, which alleviated germ cell death in the *D1* mutant (Figure 3—figure supplement 1C-D). When quantified, we observed an increased frequency of chromosome breaks (though not specific to a particular chromosome or satellite DNA) that are indicated in the text (subsection “Loss of D1/HMGA1 leads to accumulation of DNA damage”, last paragraph). In Figure 3H, we show a representative image of chromosome breaks (arrowheads) from *D1* mutant testes. We have added a more detailed description to the text (see the aforementioned paragraph) and to the revised figure legend to clarify these points.

3) The live imaging in Figure 4 should have some quantification, such as how frequently budding events were observed. Also, the experiment could be more informative if it were repeated with markers for chromatin (such as histone H2B, which is commonly used) and nuclear envelope, to reveal more details about how nuclear envelope defects lead to budding and whether chromatin is initially in the bud.

The reviewers have brought up a good point and we have addressed these comments by repeating the live imaging experiment in *Drosophila* using Df31-GFP to mark the nucleus and H2Av-RFP to label chromosomes (revised Figure 4A-B). In *D1* mutant spermatogonia, this experiment clearly demonstrates that the nucleus forms a bud, followed by chromatin leaking into it, resulting in micronuclei. In addition, we have quantified all of our live imaging in the form of total imaging time (defined as number of cells x total imaging duration), total budding time (defined as total time with budded micronuclei/total imaging duration) and number of budding events, which are now described in the Materials and methods. These data are reported in the text (subsection “Micronuclei formation in D1 mutant/HMGA1 knockdown cells is due to budding from the nucleus during interphase”) and in the figure legend for Figure 4.

4) In Figure 6 threads between chromosomes are shown. These are said to be between heterologous chromosomes in the Discussion, but we don't see data that clearly distinguish between homologous and heterologous chromosomes. If the authors have such data, they should mention it, and if not they should be clear in the Discussion that heterology is an interpretation. The model schematic (Figure 6E) could be more informative. How do the threads combine parts of different chromosomes? Do they consist of DNA from each chromosome cross-linked by D1/HMGA1 proteins? A little more detail in the schematic would help readers understand the model.

This is a very fair comment and we have edited the language to appropriately reflect the experiment and the conclusions drawn from it (subsection “D1 bundles satellite DNA from multiple chromosomes to form chromocenter”, third paragraph). In *Drosophila*, we have clearly observed AATAT-containing DNA threads linking heterologous chromosomes using FISH experiments (e.g. Figure 6C). Allied with previous studies showing satellite DNA containing threads connecting heterologous chromosomes (1–3), we are confident that these protein/DNA threads connect heterologous chromosomes. We have also updated our model (Figure 6E), showing that D1/HMGA1 physically link satellite DNA from multiple chromosomes within chromocenters.

5) What is the connection between chromocenters and nuclear envelope defects? It's not obvious why chromosomes need to be linked to keep the nucleus intact. This idea is crucial for the overall message of the paper, so it merits some discussion even if we don't know the answer yet.

We are grateful for this thoughtful comment and we agree with the reviewers that this is an exciting direction for future research. Previous reports have shown a close relationship between heterochromatin and the nuclear envelope (NE); heterochromatin/chromocenters are frequently localized along the nuclear envelope (4). Consistently, the NE is a transcriptionally repressive environment although other possible functions of this stereotypic pattern have not been discussed. Multiple studies have demonstrated that the nuclear envelope is prone to breaches, especially in response to external mechanical forces such as cytoskeletal forces and cellular migration (5–8). Therefore, we imagine that chromosome bundling in the form of chromocenter helps maintain all the chromosomes inside the nucleus in the face of mechanical challenges. In the absence of chromocenter formation (as in D1 mutant), unbundled individual chromosomes may be subjected to mechanical forces, exerting forces in multiple directions, which may tear apart the nucleus. The Discussion section of the paper has been reworked in order to incorporate this comment (third paragraph). Citations have been added where needed.

6) The first heading in the Results section states that D1 and HMCA1 bind satellite DNA, but the data do not show direct binding. The heading should probably be limited to what is directly shown by the data. Points supported by data in other papers can be explained elsewhere in the text.

D1 and HMGA1 have been well-characterized biochemically in their ability to directly bind satellite DNA (9–13). However, this is a fair comment by the reviewers and we have edited the title to “The multi-AT-hook satellite DNA binding proteins, *Drosophila* D1 and mouse HMGA1, localize to chromocenters”.

7) Do the authors have any ideas about why the D1 phenotypes are most severe in spermatogonial cells (subsection “D1 and HMGA1 are required for organizing chromocenters”, first paragraph)?

We thank the reviewers for pointing out this very interesting feature of *D1* mutation in *Drosophila*. We do not understand the cell type-specificity at this point. In our ongoing study, we have found that another chromocenter protein, exhibits a mild phenotype in gonads but a severe phenotype in imaginal discs and other diploid larval tissues. The only possibility that we can think of at this point is that chromocenter organization may be somewhat different in a cell type-specific manner, influencing the degree of phenotypic severity in different tissues. We aim to address this point in our future studies, as we understand more about chromocenter biology.

8) Can the authors speculate about why the satellite DNA that forms chromocenters is located near the centromere? For the function of linking different chromosomes together, it's not clear why centromere localization would be important.

As the reviewers have observantly pointed out, the overwhelming majority of chromocenter-forming satellite DNAs is present at pericentromeric locations on chromosomes. Although we can only speculate at this point, one possibility is the behavior of the nuclear envelope during the cell cycle. In many higher eukaryotes, the nuclear envelope disassembles during mitosis (*Drosophila* undergoes a ‘semi-open mitosis’, where the nuclear envelope undergoes extensive perforation, although they may look relatively intact at the level of light microscopy), which creates a challenge for nuclear envelope reassembly at the end of mitosis. Because kinetochores (i.e. centromeres) are pulled to a spindle pole, the centromere and nearby regions naturally come into close proximity at the end of mitosis. Thus, placing chromocenter-forming DNA (i.e. pericentromeric satellites) near the centromere makes sense in terms of re-establishing chromocenters by cross-linking many chromosomes and linking it to nuclear envelope reassembly. However, we feel that this is too speculative to be included in the main text of the manuscript at this point.

9) If chromocenter bundling proteins and pericentromeric satellite DNA are co-evolving (line 237), why does the D1 protein (from fly) also function in mouse cells where the satellite DNA sequences are different? If the fly protein can bind diverse satellite DNAs, then what does the co-evolution mean?

The reviewers have raised an important point regarding our ideas on the co-evolution of satellite DNAs and satellite DNA binding proteins. Please note that our experiments do not show that D1 (from *Drosophila*) can substitute the function of HMGA1 in mouse cells. Our result indicated that ectopically expressed D1 could bundle satellite DNA in mouse. With that said, the fact that fly D1 can bundle mouse satellite is striking, and naturally leads to a question how it is possible if satellite binding proteins and satellite DNA sequences are co-evolving. Our ongoing research suggests that chromocenter formation involves protein-protein and protein-DNA interactions (also please see response to essential revision #15). Based on this model, we speculate that a satellite DNA binding protein must evolve not only with its cognate satellite DNA but also with other satellite DNA sequences and other satellite DNA binding proteins that are important for chromocenter formation. We suggest that investigating the ability of satellite binding proteins to complement homolog function in closely related species may shed light on this fascinating topic.

10) What happens in mitosis, when chromocenters on different chromosomes should not be linked to each other? Are the D1/HMGA1 proteins removed in mitosis, or is there some other mechanism to inactivate them?

The reviewers highlight a very interesting direction for future research: clearly, chromocenters must be broken down to facilitate mitotic chromosome segregation. Consistently, we observe a significant reduction of chromosome-associated D1/HMGA1 in mitosis. Previous studies have identified that mitotic phosphorylation of HMGA1 dramatically reduces its affinity for DNA (14), which is suggestive of a cell cycle regulated mechanism for chromocenter dissolution in mitosis.

11) Chromocenters are not always as visible in some cells (e.g., human cell lines) as they are in the mouse and fly cells in this paper. Do the authors think they are less important in these cases, or still important but not as apparent by standard DNA staining for some reason?

We appreciate this insightful comment. Although chromocenters are not visible as ‘DAPI-dense foci’ in human cells, pericentromeric satellite DNA in human cells (such as HsatII) is known to be clustered into foci when visualized by FISH (15). Thus, we would suggest that human cells contain chromocenters even though they are not visible by DNA staining. This feature of human cells may reflect different abundance of pericentromeric satellite DNA, relative AT-contents, or even DNA secondary structure that might make satellite DNA refractory to DAPI staining.

12) In Figure 1 and Figure 1—figure supplement 2 the authors present data on% disruption of chromocenters, but it is not clear how disruption is defined. Is there a maximum number of foci that are considered to be not disrupted, or a minimum number of foci that are considered to be disrupted? These numbers would presumably differ between Drosophila and mouse. In most of the photos the difference seems clear, but supplement Figure 1—figure supplement 2J-K is a case where having a definition of chromocenter disruption would seem to be helpful.

We thank the reviewers for pointing this out. We observed that loss of D1/HMGA1 resulted in abnormal shape and thread-like morphology of pericentromeric satellite DNAs. Due to the thread-like appearance of satellite DNA, it is impossible to reliably estimate number of chromocenters per cell, although we stress that satellite DNA clustering is nevertheless defective. We therefore define chromocenter disruption as cells in which satellite DNA FISH adopt thread-like morphology in interphase nuclei, similar to a chromocenter scoring method utilized previously (16). This scoring method accounts for defective chromocenter formation in cells with a higher baseline of chromocenters such as C2C12 cells in Figure 1—figure supplement 2J-K. We have added this clarification to the Materials and methods section.

13) The authors' intriguing hypothesis about the function of chromocenters is welcome, but we think they should be slightly more cautious in ascribing universality to it. This should start in the title, which we think would be better as "A conserved function for chromocenters", since in Drosophila they have only shown that the AATAT satellite, not other satellites, participates in chromocenter formation. In the Introduction they say "a rationale for the very existence of pericentromeric satellite DNA is still lacking", but selfish DNA is a rationale that we think is widely accepted for some satellites. The use of a more qualified statement like " a clear rationale for the existence of most satellites is lacking" would be more accurate. On the flip side, budding yeast lacks satellites and chromocenters, yet manages to keep 16 chromosomes together, perhaps through attachment of the chromosomes to the spindle pole body. A little nuance in allowing for exceptions would not diminish the appeal of their hypothesis.

We appreciate this feedback from the reviewers and we have toned down our language in ascribing universality to satellite DNA and chromocenter function throughout the manuscript.

With regards to the title, the term ‘chromocenter’ in *Drosophila* is most frequently used to describe the one with highly polytenized chromosomes in the post-mitotic cells in salivary gland. We are afraid that using the term ‘chromocenter’ in the title may mislead the readers that our manuscript is about this specialized chromocenter, whereas our study rather highlights its importance in mitotically dividing cells, and its common characteristics with other cells (mitotically dividing mouse cells). Prior to our initial submission, we have undergone extensive discussion on the title, and we felt that ‘satellite DNA’ is better suited (not to mislead readers to salivary gland chromocenter), particularly because we study distinct satellites from two species. To avoid claiming too much universality, we decided to use ‘A conserved function….’ Instead of ‘The conserved function…’. Because we are not claiming that chromocenter formation is *the* only function of pericentromeric satellite, we hope that this title is acceptable. However, we very much welcome alternative suggestions that can further tone down implication of universality without misleading readers to salivary gland chromocenter.

14) The authors say that D1 and HMGA1 "may possess an orthologous and conserved function". Are the proteins orthologous? A little more discussion of AT hook proteins and their phylogenetic distribution in eukaryotes would be appreciated, perhaps with a supplemental cartoon of the proteins and their AT hook domains.

HMGA1 and D1 have been previously speculated to be orthologues based on their ability to bind satellite DNA and similar domain organization; they consist of N-terminal AT-hooks and a C-terminal acidic domain (9, 12, 17). However, D1 contains 10 AT-hooks whereas HMGA1 contains 3 AT-hooks, making it difficult to call them orthologous purely based on the primary protein sequences. We have bolstered this hypothesis with our data showing that D1 can localize to mouse chromocenters and enhance the clustering of mouse chromosomes. We have added slightly more detailed description on the protein primary structures in the revised text (subsection “The multi-AT-hook satellite DNA binding proteins, Drosophila D1 and mouse HMGA1, localize to chromocenters”, first paragraph).

15) In the Discussion, do they imagine that all chromocenters are based on AT hook proteins, or might there be other proteins that bundle multiple DNA strands of other satellites? A little more thought about the scope of applicability of their model might improve the Discussion.

Our current research indicates that multiple satellite DNA binding proteins function cooperatively to form chromocenters, especially in organisms with a variety satellite DNA repeats such as *Drosophila*. They do not seem to be all AT-hook proteins: instead they contain multiple DNA-binding motifs. Our speculation is that multiplicity of DNA binding domains may be the key (such that the protein can bind to multiple chromosome strands for bundling). We speculate that chromocenters are formed by protein-DNA as well as protein-protein interactions.

References:

1) Kuznetsova IS, Enukashvily NI, Noniashvili EM, Shatrova AN, Aksenov ND, Zenin VV, Dyban AP, Podgornaya OI. 2007. Evidence for the existence of satellite DNA-containing connection between metaphase chromosomes. J. Cell. Biochem. 101:1046–61.

2) Takayama S. 1975. Interchromosomal connectives in squash preparations of L cells. Exp. Cell Res. 91:408–12.

3) Burdick AB. 1976. Somatic cell chromosome interconnections in trypan preparations of Chinese hamster testicular cells. Exp. Cell Res. 99:425–8.

4) Rae MM, Franke WW. 1972. The interphase distribution of satellite DNA-containing heterochromatin in mouse nuclei. Chromosoma 39:443–56.

5) Denais CM, Gilbert RM, Isermann P, McGregor AL, te Lindert M, Weigelin B, Davidson PM, Friedl P, Wolf K, Lammerding J. 2016. Nuclear envelope rupture and repair during cancer cell migration. Science 352:353–8.

6) Hatch EM, Hetzer MW. 2016. Nuclear envelope rupture is induced by actin-based nucleus confinement. J. Cell Biol. 215:27–36.

7) Raab M, Gentili M, de Belly H, Thiam HR, Vargas P, Jimenez AJ, Lautenschlaeger F, Voituriez R, Lennon-Duménil AM, Manel N, Piel M. 2016. ESCRT III repairs nuclear envelope ruptures during cell migration to limit DNA damage and cell death. Science 352:359–62.

8) King MC, Drivas TG, Blobel G. 2008. A network of nuclear envelope membrane proteins linking centromeres to microtubules. Cell 134:427–38.

9) Rodriguez Alfageme C, Rudkin GT, Cohen LH. 1980. Isolation, properties and cellular distribution of D1, a chromosomal protein of Drosophila. Chromosoma 78:1–31.

10) Levinger LF. 1985. D1 protein of Drosophila melanogaster. Purification and AT-DNA binding properties. J. Biol. Chem. 260:14311–8.

11) Levinger L, Varshavsky A. 1982. Protein D1 preferentially binds A + T-rich DNA in vitro and is a component of Drosophila melanogaster nucleosomes containing A + T-rich satellite DNA. Proc. Natl. Acad. Sci. U.S.A. 79:7152–6.

12) Strauss F, Varshavsky A. 1984. A protein binds to a satellite DNA repeat at three specific sites that would be brought into mutual proximity by DNA folding in the nucleosome. Cell 37:889–901.

13) Radic MZ, Saghbini M, Elton TS, Reeves R, Hamkalo BA. 1992. Hoechst 33258, distamycin A, and high mobility group protein I (HMG-I) compete for binding to mouse satellite DNA. Chromosoma 101:602–8.

14) Reeves R, Langan TA, Nissen MS. 1991. Phosphorylation of the DNA-binding domain of nonhistone high-mobility group I protein by cdc2 kinase: reduction of binding affinity. Proc. Natl. Acad. Sci. U.S.A. 88:1671–5.

15) Swanson EC, Manning B, Zhang H, Lawrence JB. 2013. Higher-order unfolding of satellite heterochromatin is a consistent and early event in cell senescence. J. Cell Biol. 203:929–42.

16) Pinheiro I, Margueron R, Shukeir N, Eisold M, Fritzsch C, Richter FM, Mittler G, Genoud C, Goyama S, Kurokawa M, Son J, Reinberg D, Lachner M, Jenuwein T. 2012. Prdm3 and Prdm16 are H3K9me1 methyltransferases required for mammalian heterochromatin integrity. Cell 150:948–60.

17) Aulner N, Monod C, Mandicourt G, Jullien D, Cuvier O, Sall A, Janssen S, Laemmli UK, Käs E. 2002. The AT-hook protein D1 is essential for Drosophila melanogaster development and is implicated in position-effect variegation. Mol. Cell. Biol. 22:1218–32.